# Synthesis of sialylated human milk oligosaccharides by automated glycan assembly

Yan-Ting Kuo[1,2,3], Kim Le Mai Hoang [3] ✉ & Peter H. Seeberger [1,2] ✉

Sialic acids cap the ends of many glycan chains and play pivotal roles in cell signaling, immunity, and pathogen interactions. However, the fast synthesis of sialylated glycans by automated glycan assembly (AGA) has remained a long-standing challenge. Here we show a general strategy that leverages macro-bicyclic sialic acid building blocks to achieve reliable α(2,3)- and α(2,6)-sialylation on solid support. Using this method, a collection of nine sialylated human milk oligosaccharides (HMOs) is assembled, including fucosyldisialyllacto-*N*-tetraose (DSLNF II), a highly branched, fucosylated structure that is very difficult to synthesize by solution-phase methods. An improved global deprotection protocol provides access to pure, functionalized complex glycans suitable for further biological studies. This work provides the broadly applicable solution for automated chemical sialylation, opening the door to prepare collections of sialylated glycans for biomedical research.

Sialic acids constitute a family of over 50 structurally distinct nine-carbon nonulosonic acid sugars[1], commonly found capping the non-reducing ends of glycans on glycoproteins, glycolipids, and glycoRNAs[2]. The most prevalent form, *N*-acetylneuraminic acid (Neu5Ac), can be further modified by acetylation, methylation, or sulfation at various hydroxyl positions[3]. Through their presence on cell-surface glycans, sialic acids mediate a wide range of biological interactions, from cell differentiation and immune regulation to pathogen adhesion, and thus play central roles in processes such as cell signaling, immune responses, and infectious diseases[4–6].

Human milk oligosaccharides (HMOs), found in human breast milk, are the third most abundant solid component of milk[7] with key roles in infant nutrition, immune development, and gut microbiota modulation[8–10]. One-fifth of the over 200 distinct HMO structures are sialylated[11–14]. Access to pure molecules is crucial for elucidating their biological functions, yet natural sources yield only minute quantities that are extremely hard to isolate individually[15]. To date, studies on sialylated HMOs have focused on simple trisaccharides, such as 3′-sialyllactose (3′-SL) and 6′-sialyllactose (6′-SL), primarily due to their commercial availability[16–18]. Access to structurally defined sialylated

glycans in sufficient quantities is essential for fundamental studies and for developing glycan-based diagnostics or therapeutics.

However, obtaining homogeneously pure sialylated oligosaccharides remains challenging. Various solution-phase approaches have been explored for their synthesis, including purely chemical routes[19,20], enzymatic assembly[21–24], and chemoenzymatic methods[25,26], and all come with limitations in scope or efficiency. While purely chemical synthesis offers great flexibility, it is labor-intensive and time-consuming due to the requirement for purification after each step. Enzymatic and chemoenzymatic approaches offer complete stereo-selectivity but are often limited by the availability and substrate specificity of glycosyltransferases.

Automated glycan assembly (AGA) has emerged as a powerful technology to quickly synthesize complex oligosaccharides[27,28]. During AGA, monosaccharide building blocks are iteratively coupled on a solid support, enabling the efficient production of glycans with minimal human intervention. AGA platform bypasses intermediate purifications and automates the assembly process, significantly accelerating the production of HMOs from weeks or months to a matter of days[29]. Despite significant advances, integrating sialylation

[1]Department of Biomolecular Systems, Max-Planck Institute of Colloids and Interfaces, Potsdam, Germany. [2]Institute of Chemistry and Biochemistry, Freie Universität Berlin, Berlin, Germany. [3]GlycoUniverse GmbH & Co. KGaA, Potsdam, Germany. ✉e-mail: Kim.Lemaihoang@glycouniverse.de; Peter.Seeberger@mpikg.mpg.de

into AGA remains problematic. The unique structural features of sialic acids contribute to this challenge: the glycosylation is hindered by a tertiary anomeric center and an electron-withdrawing carboxylate at C-1 that together favor elimination over substitution. Moreover, the anomeric effect and the absence of a participating group at C-3 make controlling the stereochemistry outcome difficult, often resulting in low α-selectivity[30]. Consequently, attempts at chemical sialylation on solid supports suffered from low yields and significant 2,3-elimination side-products[31–33].

Specialized building blocks were employed to address these challenges in prior AGA strategies. Bypassing the difficult on-resin sialylation using α-sialylated galactoses (Neu5Ac-α(2,6)-Gal or Neu5Ac-α(2,3)-Gal) with the correct stereochemistry furnished the desired molecules (Fig. 1a)[34]. This method requires synthesis of customized disaccharides for each target structure because natural sialylated glycans vary in linkage positions and branching. On-resin chemical sialylation using cyclic O-4,N-5-oxazolidinone-protected sialic acid donors improved α-selectivity, but the method was effective only for mono- or disaccharide acceptors (Fig. 1b)[35]. A more general, robust solution for AGA-based sialylation remained elusive.

In macrobicyclic sialyl donors[36], the sialic acid's C-1 carboxylate is tethered to its N-5 carbamate via a finely tuned aliphatic linker to enforce α-face attack by incoming acceptors. This design both improved α-selectivity and mitigated elimination by suppressing the formation of an anti-Bredt conformation. A 2,2,2-tri-chloroethoxycarbonyl (Troc)-analog was built into the tether to enhance the donor's reactivity and provide a handle for further functionalization at the N-5 position. Inspired by this seminal work, we set out to integrate macrobicyclic sialic acids into AGA (Fig. 1c) and establish an on-resin sialylation protocol in order to access complex natural products.

In this work, after developing efficient glycosylation conditions, we achieve efficient α(2,6)- and α(2,3)-sialylations on solid support. A collection of nine sialylated human milk oligosaccharides (HMOs) of varying complexity is synthesized using the commercial Glyconeer® as well as home-built devices. Highly complex, branched structures such as fucosyldisialyllacto-N-tetraose (DSLNF II), a fucosylated and di-sialylated heptasaccharide, and a formidable challenge in solution-phase methods[37–39] are prepared. The AGA platform enables rapid exploration of protecting groups, reaction sequences, and other parameters, revealing unexpected effects of coupling order and protecting-group configuration on glycosylation efficiency. We develop an improved global deprotection protocol to cleanly remove protecting groups and furnish fully deprotected glycans suitable for biological applications. These advances highlight significant methodological breakthroughs in AGA.

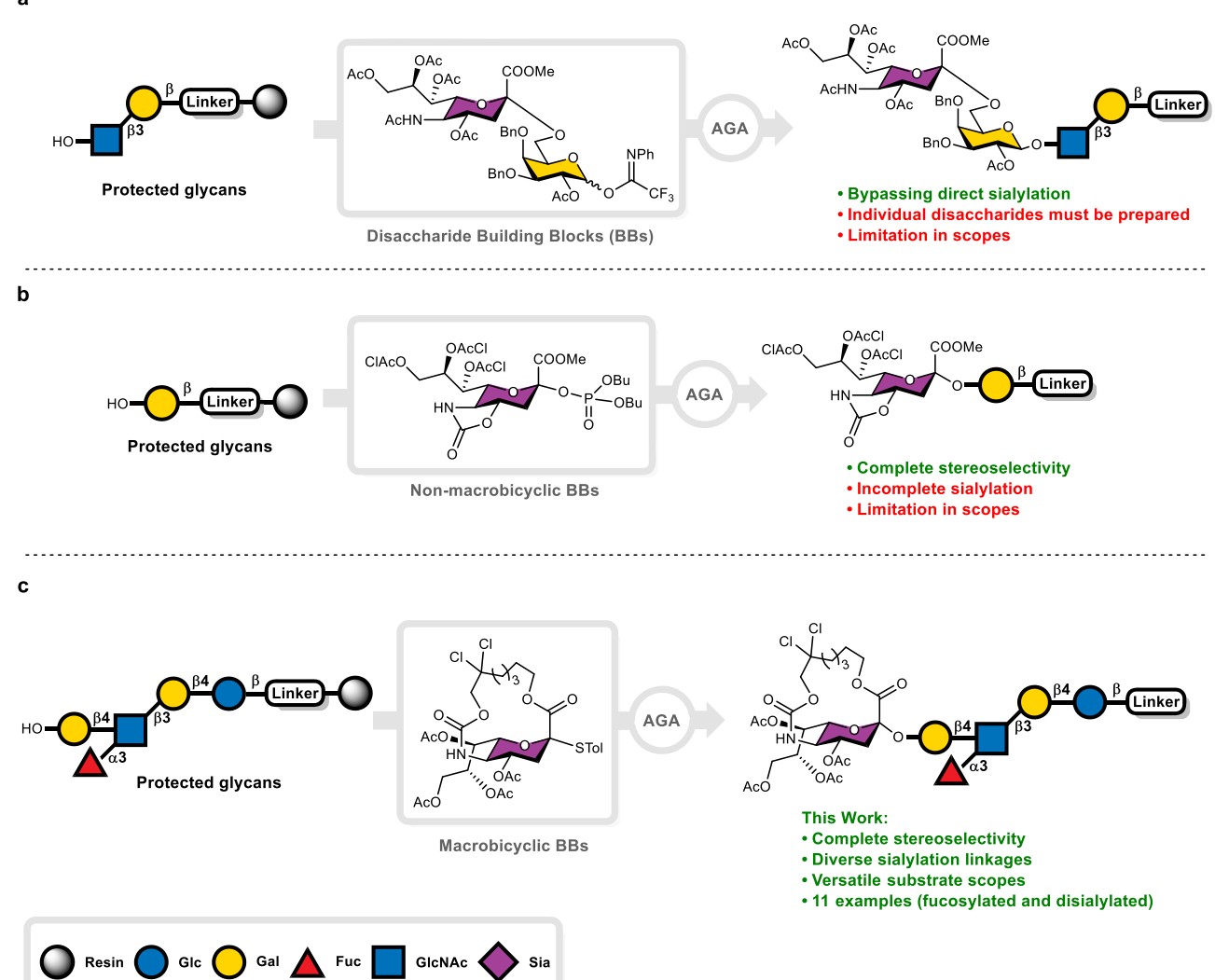

**Fig. 1 | Synthetic strategies to prepare sialylated glycans by AGA. a** Using sialylated disaccharides as building blocks. **b** Using non-macrobicyclic monosaccharides as building blocks. **c** This work: employing macrobicyclic monosaccharide building blocks.

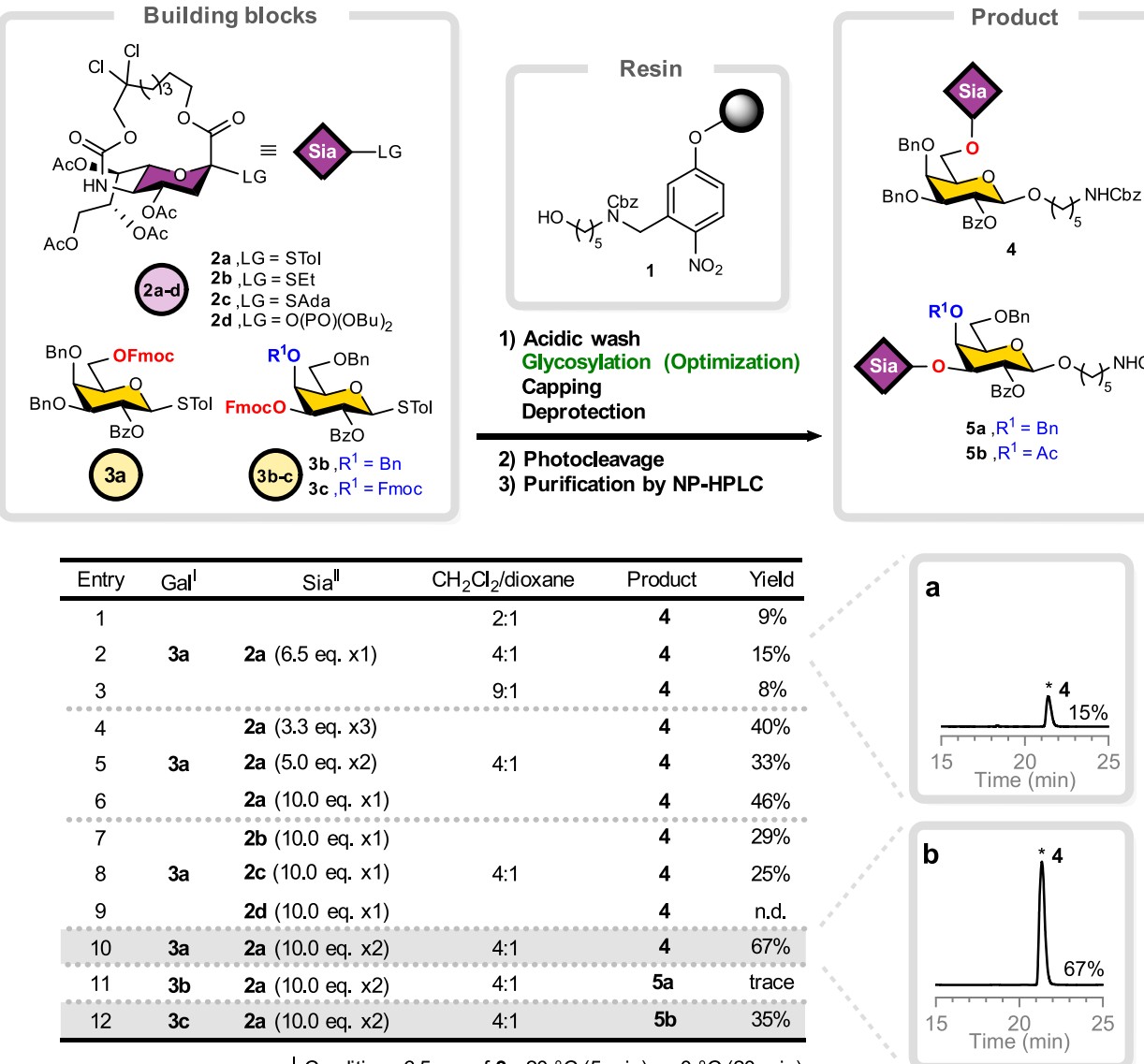

**Fig. 2 | Sialylation during AGA. a** Crude HPLC of Entry 2. **b** Crude HPLC of Entry 10. Isolated yields after NP-HPLC purification are shown; LG leaving group, Sia sialic acid building blocks, Gal galactose building blocks, n.d. not detected. Source data are provided as a Source Data file.

## Results and discussion

### On-resin optimization of sialylation (Fig. 2)

Solution-phase sialylation conditions developed by Ando et al.[36] had to be adapted to solid-phase synthesis due to the physical and chemical difference between solution- and solid-phase synthesis approaches[40]. A simple model was employed to study the on-resin α-sialylation of galactosyl acceptor **3a** using macrobicyclic donors **2a–2d**. The solvent combination used for the activation step proved critical[41]. Using a dichloromethane/dioxane mixture (4:1) for the activator solution (NIS/TfOH) gave the highest disaccharide conversion (Entry 2), whereas increasing the ratio of dioxane led to a slower reaction rate (Entry 1). Conversely, using less dioxane compromised the stability of the activator system, resulting in variable yields (Entry 3). Examining the effective concentration of sialyl donor, by varying how the ten equivalents were delivered, revealed that adding the entire ten equivalents of thiosialoside donor **2a** in a single portion gave a better outcome than multiple additions (Entry 4–6). Single-dose delivery likely maximizes the local donor concentration and the frequency of

productive collisions with the solid-supported acceptor. Under these conditions, two consecutive coupling cycles of donor **2a** afforded the sialylated disaccharide **4** in 67% isolated yield (Entry 10). Highly reactive thiosialosides **2b** and **2c** resulted unexpectedly in lower sialylated product (Entry 7 and 8), while sialyl phosphate **2d**, a popular sialyl donor in solution-phase[42–44], failed to sialylate on solid-phase (Entry 9). In the more sterically hindered environment of the polystyrene matrix, very reactive sialyl donors may not have sufficient time to reach the acceptor site before decomposing. On the other hand, lowering the reaction temperature to suppress elimination did not improve sialylation (Supplementary Fig. 1), due to reduced swelling and mixing of the resin. Isolated yields were determined for the fully purified oligosaccharides after completion of the entire workflow, including acidic wash, glycosylation, capping, deprotection, and material losses incurred during downstream purification steps such as photocleavage and preparative HPLC, relative to the theoretical maximum loading of the solid support. Because these values assume uniform efficiency across all intermediate steps, they do not directly reflect the coupling

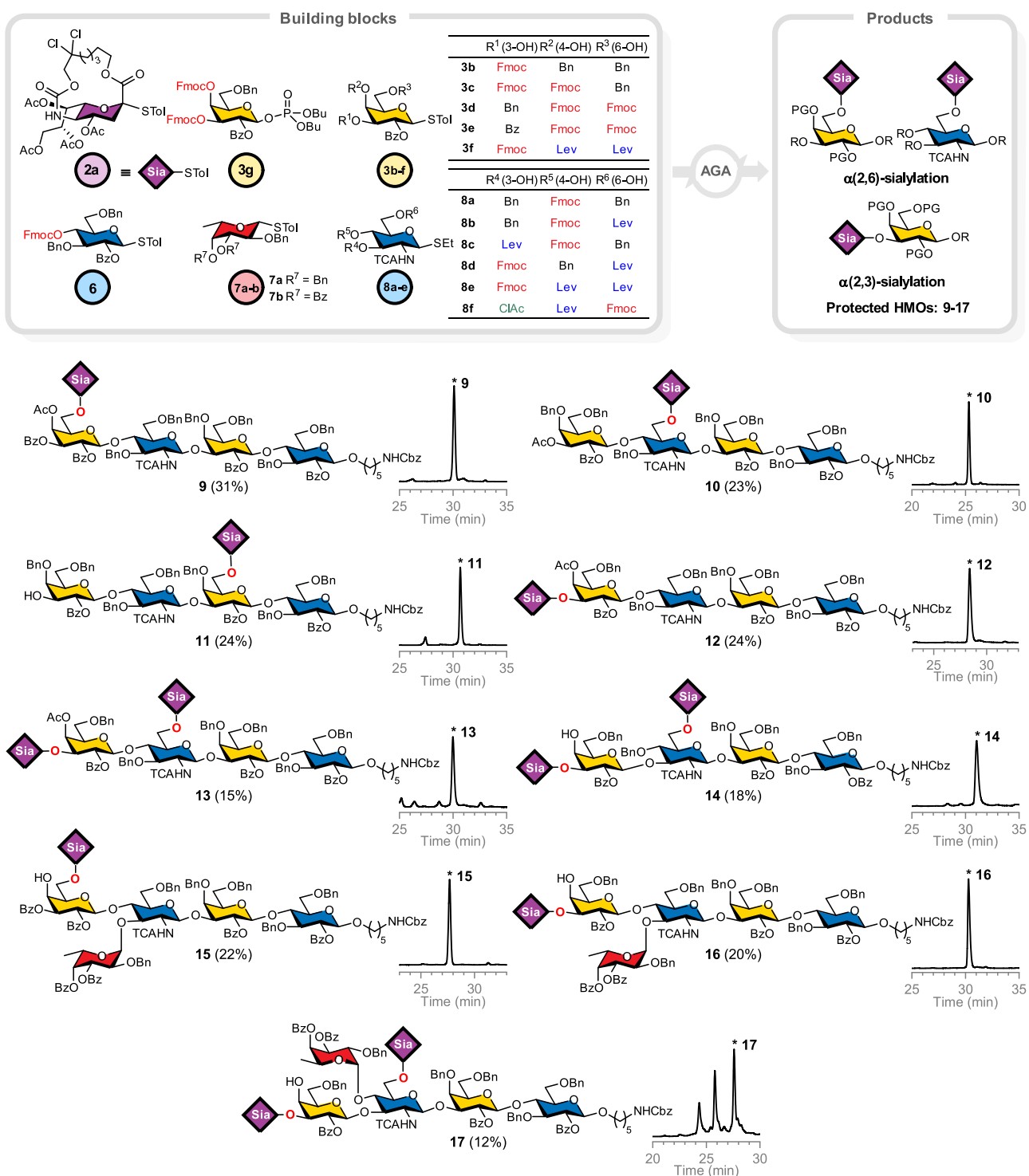

**Fig. 3 | Automated syntheses of sialylated HMOs with crude NP-HPLC traces.** Isolated yields after NP-HPLC purifications are shown in parentheses. PG protecting groups. Source data are provided as a Source Data file.

efficiency of the sialylation. Next, we turned to the more challenging α(2,3)-sialylation, where galactose acceptor **3b** carried a C-4 substituent and only trace amounts of disaccharide **5a** were detected (Entry 11), even after multiple sialylation cycles. Employing acceptor **3c** with two temporal protecting groups at C-3 and C-4 afforded disaccharide **5b** in 35% isolated yield (Entry 12). Exposure of both C-3 and C-4 hydroxyl groups during the sialylation relieves steric crowding and drastically improves coupling efficiency at C-3 without sialylation at C-4.

## Automated synthesis of sialylated HMOs (Fig. 3)

Having established effective sialylation conditions, the scope and robustness of the method were explored by synthesizing a collection of sialylated HMOs. A set of representative sialylated HMOs was targeted based on the lacto-*N*-tetraose core (LNT, Gal-β(1,3)-GlcNAc-β(1,3)-Gal-β(1,4)-Glc) and the lacto-*N*-neotetraose core (LNnT, Gal-β(1,4)-GlcNAc-β(1,3)-Gal-β(1,4)-Glc), to evaluate α(2,6)- and α(2,3)-sialylations at terminal and internal positions, as well as fucosylation in combination with sialylation. In total, nine sialylated HMO targets were

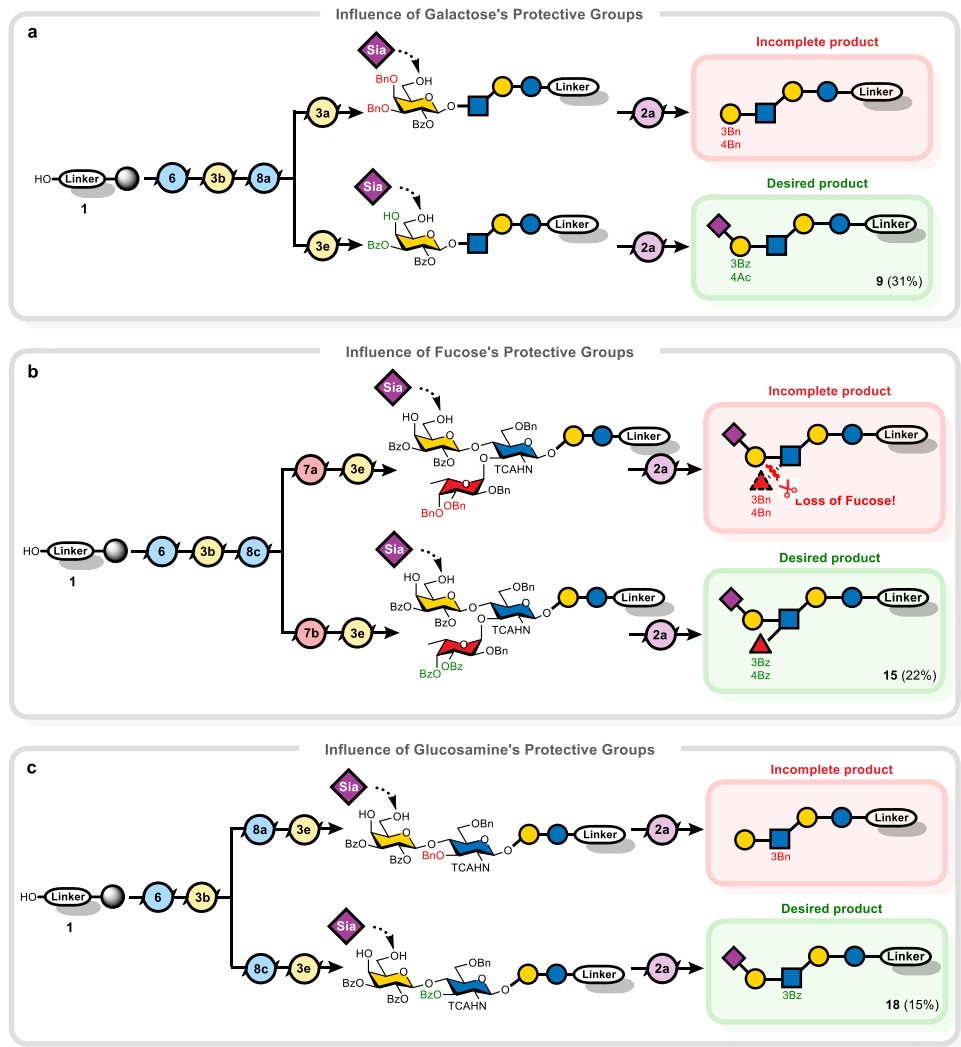

**Fig. 4 | Influence of protecting groups on remote monosaccharides on sialylation. a** Protecting groups on galactose affect α(2,6)-sialylation. **b** Protecting groups on fucose affect α(2,6)-sialylation. **c** Protecting groups on glucosamine affect α(2,6)-sialylation. Isolated yields after NP-HPLC purification are shown in parentheses.

prepared (Fig. 3), featuring biologically relevant compounds such as the influenza decoy LSTc **9**[45,46], the NEC-preventing agent DSLNT **14**[47,48], and the cancer-associated glycan DSLNF II **17**[49,50]. A set of 16 monosaccharide building blocks was prepared on a multi-gram scale. Initial efforts focused on sialylation at the non-reducing end of an LNnT core, yielding a LSTc pentasaccharide **9**. Sialyl building block **2a** was appended to the terminal galactose of LNnT. Using conventional 6-O-Fmoc protected thiogalactoside **3a**, sialylation failed even after two cycles and produced only tetrasaccharide (Fig. 4a). This result was surprising because the same galactose acceptor was sialylated in model studies (Fig. 2). Swapping mono-Fmoc thiogalactoside **3a** for dual-Fmoc thiogalactoside **3d** or **3e** afforded modest yields of sialylated **9** and **S2** (Supplementary Fig. 2). The nucleophilicity of the terminal galactose was dramatically reduced, whereas the electronic influence of C-3 substitution was negligible. The results suggested that when the same galactose moiety is part of a larger oligosaccharide, such as the LNnT core, its reactivity is significantly attenuated by the cumulative electronic and steric effects of the neighboring sugars. Next, the number of sialylation cycles was varied in an attempt to improve the yield (Supplementary Fig. 3). Up to four sialylation cycles yielded 15–31% of mono- and di-sialylated oligosaccharides **9–14** (Fig. 3). While these yields are moderate, the success showcases the ability of the AGA platform to conduct multiple sequential sialylations

on one molecule. Just two sialylation cycles were sufficient to operate on an internal glucosamine, in contrast to both terminal and internal sialylation of galactose.

Fucosylation of sialylated HMOs is required to synthesize Lewis[x]-containing structures. The α(1,3)-linked fucose on *N*-acetyl glucosamine (GlcNAc) can potentially influence the outcome of adjacent glycosylations, due to steric or electronic effects, and served to test the method.

Two fucosylated hexasaccharides served as targets: terminal α(2,6)-sialic acid-containing **15**, and **16** bearing a terminal α(2,3)-sialic acid. The strategy was to build the core with a glucosamine building block **8c** bearing a temporal C-3-levulinyl (Lev) group, to be later unveiled selectively for fucosylation, followed by galactosylation at C-4 and final sialylation. Two fucose building blocks were examined: **7a**, carrying C-3 and C-4 benzyl ethers (Bn), and **7b**, containing benzoyl esters (Bz) at those positions (Fig. 4b). Using thiofucoside **7a** failed to furnish any fucosylated final product. Detailed investigations revealed that fucosylation was indeed completed, but the glycosidic bond between fucose and glucosamine had been severed at some point later in the synthesis. The acid-sensitive *O*-fucosidic bond was inadvertently cleaved during the final α(2,6)-sialylation step in the presence of both NIS/TfOH and the activated sialyl donor. With thiofucoside **7b**, the fucosidic linkage survived, and the desired protected hexasaccharide

**15** was obtained in 22% yield. The benzoyl esters on fucose likely improved its stability during sialylation and prevented its premature loss[51,52].

During the synthesis of hexasaccharide **15**, it became evident that the appendage at the C-3 position of the internal glucosamine can exert a long-range influence on the efficiency of terminal α(2,6)-sialylation on galactose. To probe this effect, we synthesized a series of LNnT-based tetrasaccharides carrying either a benzyl ether or benzoate ester (Fig. 4c). Both tetrasaccharides were subjected to α(2,6)-sialylation (only two cycles of sialyl donor **2a**) on the terminal galactose. While the presence of 3-O-Bn glucosamine resulted in only partial sialylation, needing at least two additional cycles, the presence of 3-O-Bz glucosamine resulted in complete sialylation in just two cycles, similar to fucosylation (Supplementary Fig. 5). This observation suggested that the electron-withdrawing benzoyl group, and likewise fucose, might influence the conformation of the galactose-glucosamine linkage or the electron density of galactose, effectively reducing steric hindrance or otherwise making the galactose terminus a better acceptor for α(2,6)-sialylation. On the other hand, during the synthesis of terminal α(2,3)-sialylated hexasaccharide **16**, the presence of fucose did not measurably improve the efficiency, as still four cycles of sialylation were required to install the α(2,3)-sialyl linkage. Apparently, fucose accelerates only α(2,6) sialylation. The modular and flexible nature of AGA allows for rapid exploration of protecting group and substituent effects, yielding insights that would be difficult to obtain otherwise.

### Automated synthesis of DSLNF II (Fig. 5)
DSLNF II is a complex HMO with a branched LNT core bearing one α(1,4)-linked fucose and two sialic acids. This heavily branched structure makes it a formidable synthetic target. Enzymatic approaches failed to produce DSLNF II[38,39], leaving Kiso's total synthesis the only means to access the material[37]. AGA of DSLNF II would demonstrate its capability to quickly build even very challenging glycans. The initial strategy (Route A) aimed to α(2,6)-sialylate the internal glucosamine early, prior to fucose and galactose extension. Using glucosamine building block **8e** (3-O-Fmoc, 4,6-di-O-Lev), a trisaccharide core was assembled. At this stage, two options were explored. After removing both Lev groups to expose the 4- and 6-hydroxyl groups, α(2,6)-sialylation occurred exclusively at the primary acceptor[37,53], followed by α(1,4)-fucosylation at 4-OH (Route A1). The early sialylation and fucosylation on glucosamine were successful, but the subsequent addition of the branch galactose after 3-O-Fmoc removal failed. The 3-OH acceptor was sterically blocked by the adjacent fucose and sialic acid moiety. Alternatively, Gal-β(1,3) was first introduced after 3-O-Fmoc removal, followed by α(2,6)-sialylation and α(1,4)-fucosylation (Route A2). This attempt also failed as the final fucosylation conditions unexpectedly cleaved the GlcNAc linkage to fragment the molecule.

Given the difficulties with late-stage fucosylation and galactosylation, both sialic acids were placed at the very end, a late-stage bis-sialylation approach (Route B). New glucosamine building block **8f** (3-O-chloroacetyl, 4-O-Lev, 6-O-Fmoc) was prepared so that each hydroxyl group could be unmasked independently. Removal of 4-O-Lev for 4-O-fucosylation, followed by chloroacetyl removal for 3-O-galactosylation prior to 6-O-sialylation (Route B1) did not meet with success as the chloroacetate was not fully cleaved due to the neighboring fucose moiety, and instead cleavage of the glucosamine linkage was observed (Supplementary Fig. 7). Success was achieved in Route B2 by early chloroacetate removal to append galactosyl phosphate **3g** at C-3, followed by fucosylation at C-4 and di-sialylation at both internal glucosamine and terminal galactose acceptors. The use of galactosyl phosphate **3g** in place of thiogalactoside **3c** was necessary to ensure β(1,3)-Gal (Supplementary Fig. 8).

Desired product **17** (12% yield) was accompanied by a small amount of mono-sialylated product, indicating that the internal glucosamine is the most difficult to sialylate. Attempts to force the sialylation with additional cycles resulted in decomposition. Overall, the automated synthesis of DSLNF II showcases how AGA, combined with strategic building block and sequence design, can surmount formidable challenges in oligosaccharide assembly.

### Global deprotection of protected sialylated HMOs (Fig. 6)
With protected sialylated oligosaccharides in hand, the final step was a gentle deprotection that preserved sensitive glycosidic linkages. Both sialic acid and fucose are acid/base-labile, and the macrobicyclic tether carries a base-labile, Troc-like carbamate that is typically removed in the presence of excess zinc nanopowder and microwave heating[54,55]. We sought a unified deprotection sequence to remove benzyl/benzoyl/acetate/Troc-like/TCA/Cbz protecting groups while retaining sialyl and fucosyl integrity. Using pentasaccharide **9** (Fig. 6a), a base-first route (LiOH, 0.1 M, THF/MeOH/H₂O, 40 °C)[56], followed by Ac₂O/Et₃N capping of the freed amine and Pd(OH)₂/C hydrogenolysis in t-BuOH/H₂O, delivered only 27% of pentasaccharide **20**. Higher m/z byproducts arise likely from incomplete Troc/TCA removal and/or base-promoted peeling during prolonged exposure (Supplementary Fig. 9).

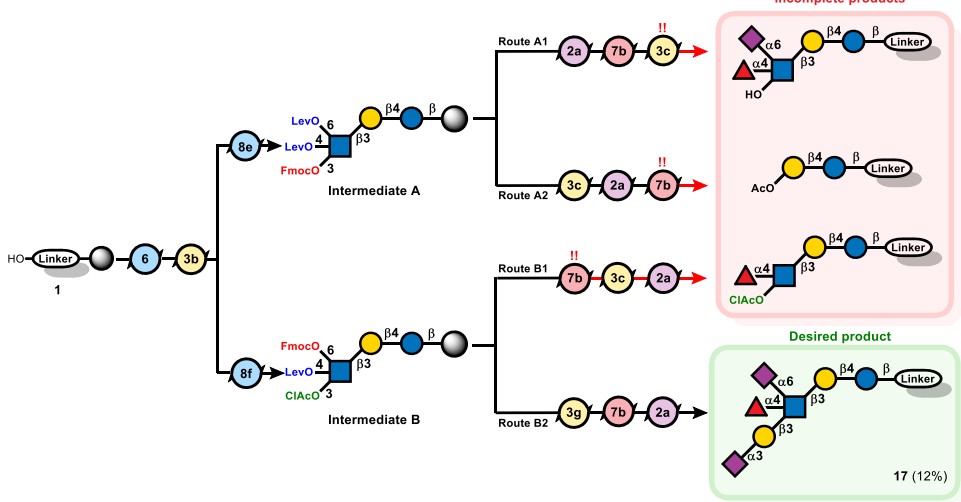

**Fig. 5 | AGA of heptasaccharide 17 via different synthetic sequences.** Isolated yields after NP-HPLC purification are shown in parentheses.

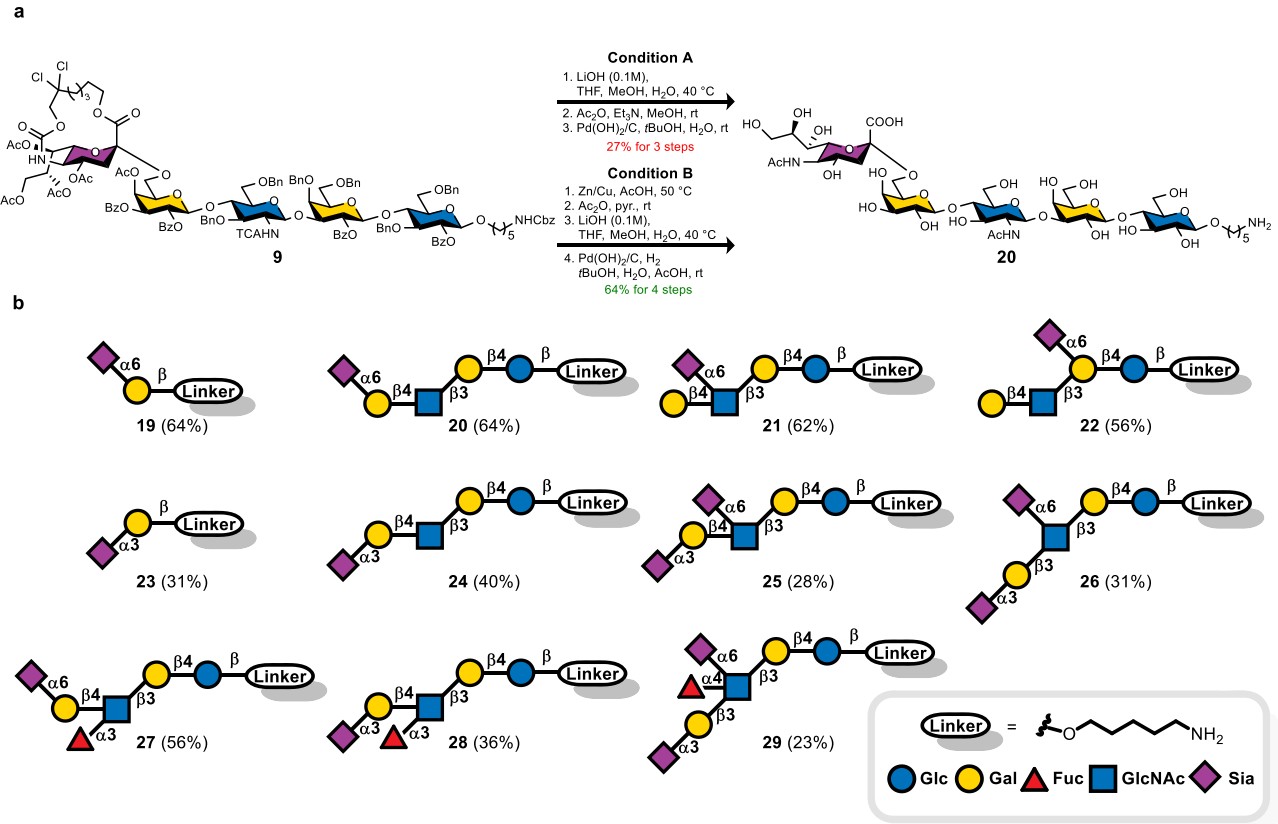

**Fig. 6 | Global deprotection of sialylated glycans. a** Optimization of global deprotection. **b** Sialylated disaccharides and HMO products. Isolated yields after RP-HPLC purification are shown in parentheses.

Gratifyingly, a reductive start using Zn-Cu couple in AcOH at 50 °C cleanly cleaved the N-5 Troc-like carbamate and fully reduced NHTCA to NHAc, thereby circumventing major complications owing to poor solubility and incomplete reduction of TCA groups during the hydrogenolysis step[57]. After acetylating the nascent amino group with Ac$_2$O/pyridine, mild saponification with 0.1 M LiOH removed all esters, and a final Pd(OH)$_2$/C hydrogenolysis in $t$-BuOH/H$_2$O/AcOH removed the remaining benzyl/Cbz groups. The use of acetic acid as a cosolvent helped to suppress base-mediated sialic cleavage and aided stubborn benzyl removal. This sequence furnished product **20** in excellent yield and purity (64% over four steps). Subsequently, mono α(2,6)-sialylated disaccharide **19**, pentasaccharides **20**–**22**, and hexasaccharide **27** were obtained in 56–64% overall yields. Oligosaccharides **23**–**26** and **28**–**29** bearing α(2,3) sialic acids showed reduced recovery (23–40%).

Starting from solid-support **1**, all glycans carried a 5-aminopentanyl spacer at the reducing-end terminal, providing a handle for bioconjugation, such as direct printing on NHS-activated slides for glycan arrays, or coupling to various carriers for immunogenicity studies[27]. The Zn-Cu-initiated, acetate-buffered sequence proved broadly applicable, delivering biologically relevant sialylated glycans in ready-to-use form.

In conclusion, we have established a broadly applicable solution to one of the long-standing challenges in automated glycan assembly: the on-resin construction of sialylated oligosaccharides. Through integration of a macrobicyclic sialic acid donor into the AGA workflow and systematically optimizing key parameters, we achieved efficient α(2,6) and α(2,3) sialylations on a solid-phase automated platform. The requirement for excess bicyclic donors is a current limitation; however, the excess donors can be recovered and reactivated/recycled, which helps mitigate material use. The power of this approach is demonstrated by the AGA of nine complex HMO structures, including fucosylated and multiple sialylated examples such as DSLNF II, a highly branched glycan that was previously beyond the reach of enzymatic methods. These case studies revealed critical insights into glycosylation sequence and how effects such as the order of assembly and the choice of protecting groups on remote monosaccharide constituents can dramatically influence sialylation. These insights will guide future synthetic planning in complex glycan assembly. An improved deprotection protocol enabled the clean removal of even the most stubborn protecting groups, delivering analytically pure glycans.

Together, these findings significantly expand the capabilities of AGA. Homogenous sialylated oligosaccharides can now be accessed in a rapid and modular fashion. AGA helps to probe the effect of fucose or a protecting group at a remote position and offers a powerful platform for discovering structure-activity relationships in synthetic carbohydrate chemistry.

## Methods
### General
The synthesis and characterization of monomer building blocks along with the synthetic protocols and HPLC chromatograms for AGA of protected-oligosaccharides (**4, 5a-b** and **9**–**18**) and deprotected-oligosaccharides (**19**–**29**) are provided in Supplementary Information. [1]H, [13]C, and two-dimensional (2D) NMR spectra of the compounds described in the article are available in Supplementary Information.

### AGA
The automated synthesizer executes a series of commands that are combined into modules to achieve specific transformations (see Supplementary Information).

## Acidic wash module

Once the temperature of the reaction vessel has adjusted to the desired temperature of −20 °C by the cooling device, 1 mL of the Acidic Wash Solution is delivered to the reaction vessel. After three minutes, the solution is drained. Finally, the resin is washed with 3 mL CH$_2$Cl$_2$ (bubbling = 15 s) and drained.

## Glycosylation using the thioglycoside module

Upon draining the CH$_2$Cl$_2$ in the reaction vessel, 1 mL of Building Block Solution containing the appropriate building block is delivered from the building block storing component to the reaction vessel. After the temperature reaches the desired temperature ($T_1$), Thioglycoside Activator Solution (1 mL) is delivered to the reaction vessel from the respective activator storing component to the reaction vessel. The glycosylation mixture is incubated for the selected duration ($t_1$) at the desired $T_1$, then the reaction temperature is linearly ramped to $T_2$. Once $T_2$ is reached, it is maintained, and the reaction mixture is incubated for an additional time ($t_2$). Once the incubation time is finished, the reaction mixture is drained, and the resin is washed with CH$_2$Cl$_2$ (1 × 3 mL for 15 s), then dioxane (1 × 3 mL for 15 s), and finally CH$_2$Cl$_2$ (2 × 3 mL for 15 s).

## Glycosylation using the glycosyl phosphate module

Upon draining the CH$_2$Cl$_2$ in the reaction vessel, 1 mL of Building Block Solution containing the appropriate building block is delivered from the building block storing component to the reaction vessel. After the temperature reaches the desired temperature ($T_1$), Phosphate Donor Activator Solution (1 mL) is delivered to the reaction vessel from the respective activator storing component to the reaction vessel. The glycosylation mixture is incubated for the selected duration ($t_1$) at the desired $T_1$, then the reaction temperature is linearly ramped to $T_2$. Once $T_2$ is reached, it is maintained, and the reaction mixture is incubated for an additional time ($t_2$). Once the incubation time is finished, the reaction mixture is drained, and the resin is washed with CH$_2$Cl$_2$ (3 × 3 mL for 15 s).

## Pyridine wash module

The resin is washed with DMF (2 × 3 mL for 15 s). Then, the pre-capping solution (2 mL) is delivered into the reaction vessel and incubated for three minutes. The resin is then washed with CH$_2$Cl$_2$ (3 × 3 mL for 15 s).

## Capping module

The resin is washed with DMF (2 × 3 mL for 15 s). Then, the pre-capping solution (2 mL) is delivered into the reaction vessel and incubated for three minutes. The resin is then washed with CH$_2$Cl$_2$ (3 × 3 mL for 15 s). Upon washing, Capping Solution (4 mL) is delivered, and the temperature is adjusted and maintained at 25 °C. The resin and the reagents are incubated for 20 min. The solution is then drained from the reactor vessel, and the resin is washed with CH$_2$Cl$_2$ (3 × 3 mL for 15 s).

## Fmoc deprotection module

The resin is first washed with DMF (3 × 3 mL for 15 s), and then Fmoc Deprotection Solution (2 mL) is delivered to the reaction vessel. After five minutes, the reaction solution is drained, and the resin is washed with DMF (3 × 3 mL for 15 s) and CH$_2$Cl$_2$ (3 × 3 mL for 15 s). After this module, the resin is ready for the next glycosylation cycle.

## Lev deprotection module

The resin was washed with DMF (3 × 30 s), and DCM (1.3 mL) was added to the reaction vessel. Lev Deprotection Solution (0.8 mL) was added to the reaction vessel, and the temperature was adjusted to 25 °C. After 30 min, the reaction solution was drained, and the entire cycle was repeated twice more. After Lev deprotection was completed, the resin was washed with DMF (3 × 3 mL for 15 s), THF (3 × 3 mL for 15 s), and CH$_2$Cl$_2$ (3 × 3 mL for 15 s).

## ClAc deprotection module

The resin is first washed with CH$_2$Cl$_2$ (3 × 3 mL for 15 s), then ClAc Deprotection Solution (2 mL) was delivered to the reaction vessel. The temperature of the reagents inside the reactor vessel is then adjusted to 70 °C. After 20 min, the reaction solution is drained from the reactor vessel. The resin is washed with DMF (3 × 3 mL for 15 s). Then, fresh ClAc Deprotection Solution (2 mL) is delivered, and the process is repeated twice more. Then, the resin is washed with DMF (3 × 3 mL for 15 s) and CH$_2$Cl$_2$ (5 × 3 mL for 15 s).

## Photocleavage and normal phase purification

**Resin cleavage.** Vapourtec® E-Series UV-150 Photoreactor Flow Chemistry System was used. The medium-pressure mercury lamp was filtered by the commercial long-pass filter (No.3, red), adjusted to 80% power, and maintained at 20 °C by a cold nitrogen flow (dried ice reservoir). The resin was suspended in CH$_2$Cl$_2$ (around 10 mL) and loaded into a PTFE syringe. The suspension was pushed by a syringe pump (PHD2000, Harvard Apparatus) at 1 mL/min into the reactor coil (10 mL, 1/8-inch o.d. FEP tubing) inside the UV-chamber. After 15 min, the resin was pushed out by CH$_2$Cl$_2$ (2 mL/min). The blackened resin was filtered, and the reactor was washed with CH$_2$Cl$_2$ (10 mL). The solvent was evaporated *in vacuo*.

**Analytical NP-HPLC.** The crude product was dissolved in 4 mL of ethyl acetate (EtOAc) and analyzed using analytical HPLC (Agilent 1200 Series system). A YMC-Diol-300-NP column (150 mm × 4.6 mm I.D.) was used with a flow rate of 1.00 mL/min.

**Preparative NP-HPLC.** The crude product was dissolved in a 1:1 mixture of hexane and EtOAc (2 mL) and was analyzed on an Agilent 1200 Series system. A YMC-Diol-300-NP column (150 mm × 20 mm I.D.) was used with a flow rate of 15.00 mL/min.

## General deprotection procedure

To a suspension of protected glycan in AcOH (5 mL) was added Zn/Cu (500 mg) and stirred at 50 °C overnight. The suspension was filtered, and the solvent was evaporated *in vacuo*. Then, the crude compound was dissolved in Ac$_2$O/pyridine (4 mL, 1:1) and stirred overnight. Upon the completion of acetylation, the solvent was evaporated *in vacuo*. The acetylated crude compound was dissolved in THF/MeOH/H$_2$O (5 mL, 2:2:1), and LiOH (12 mg) was added. The mixture was stirred at 40 °C overnight. Amberlite IR-120 (H$^+$ form) was then added to neutralize. After neutralization, the reaction mixture was filtered, and the solvent was evaporated *in vacuo*. The crude compound was used for hydrogenolysis without further purification. The crude compound was dissolved in $^t$BuOH:H$_2$O (3:1, 4 mL) with AcOH (0.1 mL). The suspension was added Pd(OH)$_2$/C (10%) purged with N$_2$ and H$_2$ for ten minutes. The suspension was stirred under a H$_2$ balloon overnight. The insoluble material was removed by a CHROMA-FIL ®Xtra, RC 0.45 syringe filter. The solid was washed once with MeOH and several times with water. The filtrate was collected and concentrated *in vacuo*.

## Reverse-phase purification

**Analytical RP-HPLC.** The crude product was dissolved in 3 mL of H$_2$O and analyzed using analytical HPLC (Agilent 1200 Series system). A Hypercarb column (150 mm × 4.6 mm I.D.) was used with a flow rate of 0.7 mL/min, or a Synergi Hydro RP18 column (250 mm × 2.6 mm I.D.) was used with a flow rate of 1.0 mL/min.

**Preparative NP-HPLC.** The crude product was dissolved in 3 mL of H$_2$O and analyzed on an Agilent 1200 Series system. A Hypercarb column (150 mm × 10 mm I.D.) was used with a flow rate of 3.0 mL/min, or a Synergi Hydro RP18 column (250 mm × 10 mm I.D.) was used for analytical RP-HPLC with a flow rate of 4 mL/min.

**Reporting summary**

Further information on research design is available in the Nature Portfolio Reporting Summary linked to this article.

## Data availability

Unless otherwise stated, all data supporting the results of this study can be found in the article, Supplementary Information, and source data files. Source data are provided with this paper.

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

## Acknowledgements

We gratefully thank Olaf Niemeyer and Eva Settels for their support in NMRs, HPLCs, and other analytical devices. This work is supported by the generous financial support of the Max-Planck Society. Y.-T.K. acknowledges financial support from the European Union's Horizon 2020 research and innovation program (GLYTUNES) under the Marie Skłodowska-Curie grant agreement No 956758.

## Author contributions

K.L.M.H and P.H.S. conceived and supervised the experiments and contributed to manuscript preparation. Y.-T.K. performed the synthetic work, conducted data analysis, and prepared the initial draft of the manuscript.

## Funding

## Competing interests

Y.-T.K. declares no competing interests. K.L.M.H. and P.H.S. declare a competing interest in GlycoUniverse GmbH & Co. KGaA, the company that commercializes the Glyconeer® automated synthesizer and glycan products, including all building blocks used in this study.
