## [Transparent Peer Review file · Nature Communications]

Synthesis of Sialylated Human Milk Oligosaccharides by Automated Glycan Assembly

Corresponding Author: Professor Peter Seeberger

Version 0:

Reviewer comments:

Reviewer #1

(Remarks to the Author)

The manuscript by Seeberger and colleagues reports the synthesis of sialylated HMOs using an automated glycan assembly platform. The authors utilize macrobicyclic thiosialoside building blocks to construct $\alpha(2,3)$ - and $\alpha(2,6)$ -sialosides on solid support and demonstrate the successful synthesis of nine sialylated HMOs, including the highly branched and challenging fucosyldisialyllacto-N-tetraose (DSLNF II). However, I believe there is not enough novelty disclosed in the manuscript that would appeal to a broad readership. The advances reported do not go beyond the group's prior contributions, and the conceptual or methodological insights are not sufficiently differentiated to engage the broad readership of Nature Communications. As such, I do not believe the manuscript, in its current form, meets the threshold for publication in this journal. The major concerns are as follows:

1. The macrobicyclic sialic acid donors employed in this study have already been reported and extensively characterized by Kumura and co-workers in solution-phase synthesis. Simply transferring these previously established donors onto the AGA platform does not constitute sufficient novelty to attract the interest of Nat. Commun. readers.
2. The yields reported for the sialylation steps remain low, despite the use of 20, 40, or even more equivalents of the sialic acid donor. Such high donor loadings with modest conversion raise concerns about the efficiency and robustness of the method.
3. I am also curious whether, utilizing similar resources, the overall yields could be improved by employing disaccharide glycosyl donors that already contain the sialyl moiety. A brief discussion or comparison of this strategy would be helpful to contextualize the efficiency of the current approach.
4. On page 2, left column, the authors state: "Solution-phase sialylation conditions developed by Ando et al. had to be adapted to solid-phase synthesis due to the physical and chemical difference between solution and solid phase synthesis approaches²⁶." It is unclear which specific reference this statement is citing. The authors should clarify the corresponding citation.
5. Page 2, left column "Conversely, using too little dioxane compromised the stability of the activator system, resulting in variable yields (Entry 1-3)." The statement is misleading since the dioxane ratio is not too little in entries 1-3, its changing significantly in these three entries.
6. Page 2, right column "A set of representative sialylated HMOs was targeted based on the lacto-N-tetraose (LNT) and the lacto-N-neotetraose (LNnT) tetrasaccharides core, to evaluate $\alpha(2,6)$ - and $\alpha(2,3)$ -sialylations at terminal and internal positions, as well as fucosylation in combination with sialylation". Please refer to a scheme or provide a generic schematic representation for LNT or LNnT for the readers convenience.
7. What are the isolated yields for the individual sialylation steps in the preparation of the mono- and disialylated oligosaccharides 9–14?
8. Is the overall yield of DSLNF II by AGA is significantly high compared to Kiso's total Synthesis?

Reviewer #2

(Remarks to the Author)

This manuscript describes an automated solid-phase synthesis of sialylated HMOs.

In the early stages of this approach, sialic acid donors were commonly used in the form of pre-linked disaccharides such as Sia($\alpha 2 \rightarrow 3$)Gal NPhTFAI or Sia($\alpha 2 \rightarrow 6$)Gal NPhTFAI donors (10.3762/bjoc.8.183). However, this strategy required a distinct sialyl-galactoside building block per each linkage type and sugar "($\alpha 2 \rightarrow x$)sugarx", besides is costly for SPOS workflows

where excess of building blocks are typically needed. Moreover, the key sialylation step occurred off-resin, under non-automated manner, limiting reproducibility and time efficiency.

On-resin synthesis of sialylated oligosaccharides offers several advantages:

(1) it enables full mechanization/automation of the glycosylation step and forget about operator training, and (2) it allows the use of a single, universal sialic acid donor compatible with different substrates and linkages. This approach is particularly appealing given that sialic acids naturally occur predominantly at terminal positions, making large-scale preparation of a universal donor economically interesting.

In this context, the authors previously explored sialic acid building blocks bearing modifications at O4/N5, O9, and incorporating a dibutylphosphate leaving group (10.3762/bjoc.11.69). While these donors showed high stereocontrol, the authors argued that the applicability is yet limited in scope, being mainly effective for mono- and disaccharide targets. In the same direction, the current study evaluates a single bicyclic sialic donor for integration into the AGA workflow. The authors demonstrate its universality through the synthesis of 11 sialylated products, including nine HMOs of high biological relevance (sialylation of fucosylated and non-fucosylated lacto-N-tetraose and lacto-N-neotetraose cores). Sialylation was successfully performed at O3 and O6 of Galp, and at O6 of GlcN linkages. All products thoroughly characterized and supported by solid structural, spectroscopical and stereochemical data.

It is worth mentioning that the bicyclic donor bears an auxiliary protecting group that tethers C1–N5, which does not appear to be costly to make, and can be introduced/removed without significant difficulty.

Finally, they improved a global deprotection protocol for the post automated synthesized products.

However, I have many comments and concerns that feel need to be addressed. Please see below. But overall, I recommend publication after revisions to clarify and better highlight the major findings for the readers.

_Body of the Manuscript

Comments:

Page2, left column. Add reference after "Ando et al". (Probably 10.1126/science.aaw4866)

Fig 3. Right to the top: Please double check if products should include Sia ($\alpha 2 \rightarrow 6$) GlcN structure.

Fig 3. Structure of fucose 7a is missing.

Page 5. Left column, Automated synthesis of DSLNF II. Reference routes A1, A2 and B1, in the text.

Concerns:

The excess equivalents of the bicyclic Sia donor that were needed show a weakness that this approach usually presents, and I think the authors could state this more explicitly. Ando's stereoselective glycosylations in solution are excellent (83–99%) using just a 1.0:1.0 donor/acceptor ratio (10.1126/science.aaw4866). It is not clear to me if the yields obtained by the authors are inherent to sialylations performed in solid phase, or if Ando's solution-phase glycosylations cannot be compared directly, because the efficiency of the cleavage of the sialylated products from the solid support either decomposes the sialoside or makes the cleavage more difficult. Could the authors please make comment on this?

It is understood that the yields shown in the tables and in all the Figures correspond to the moles of product obtained after purification, once the complete automated cycle of glycosylations, capping, tPG removal, and sialylations has been completed, and that this includes the post-automation cleavage process. With this in mind, I would like to emphasize that the yields reported cannot be directly or specifically associated with the actual coupling efficiency of the sialic acid. Such an association could only be made if the authors mentioned at least that they are assuming that all oligosaccharides, regardless of size or whether they are fuco or sialylated or not, cleave within a similar range. Therefore, I suggest to the authors mention this point somewhere in the manuscript in order to guard the analysis, towards the analysis & interpretation of the reported yields. Readers should acknowledge there are so many possible confounding variables when a tremendous complex work is undergoing that some supposition has to be made at the time of analyzing outcomes, likewise it is traditionally done by solution-phase carbohydrate chemists.

Page 4, left column. Whole paragraph that starts with "During the synthesis of hexasaccharide 15" until ends first paragraph, left column Page 5. This reviewer does not feel confident about the weight given to the discussion of the influence of the substituent at position 3 of GlcN on the yields. This is because it is not simply the variation of the substituent, or even the change of the GlcN BB (8a or 8c); rather, the order of incorporation of the Galp is also being changed, and in addition there are tPG replacement reactions occurring. This concern is associated with the Conclusions section, fourth sentence that relates sialylation with assembling order.

Page 5. Left column. Automated synthesis of DSLNF II. "...($\alpha 2,6$) sialylation occurred exclusively at the primary acceptor". I assume this is expected by the authors, but I am not sure if that regioselectivity will be clear for a broader range of readers. Please, can you provide a reference that discuss this matter?

_Supplementary Material document:

Comments

S6: Fig S5. Titles (A) and (C), and corresponding left structures, 9 and 18 might be reviewed. Compound 9 is a 3OBn protected GlcN. Compound 18 has a 3OBz on GlcN.

S11: Synthesis of S5. Ester reduction to alcohol shouldn't be with aluminum based hydrate (LiAlH₄) instead of NaBH₄? Please double check.

S12: Protected sialic acid chemical structure. Should say compound S7 instead of S6. S6 is SM.

S40: Levulinic anhydride synthesis. Please double check if no base was required here.

S43: Synthesis of S16. Last paragraph: please double check if should say S16, instead of S15.

S44: Synthesis of 8f. First paragraph: please double check if should say S16, instead of S15.

S46: Please disclose the type and amount of resin used (might be mentioned elsewhere but this reviewer couldn't find it).

S55: All NMR spectra of samples prepared in CDCl₃ were collected at 50 oC, is there any specific/special reason behind?

S72: Please double check if capping module is included or not during the Gal3b coupling. Due to Sialo lacto-N-neotetraose

11 has a 3OH on terminal Galp, final capping might not be included.

S86: Please double check if capping module is included or not during the Sia2a coupling. Due to Di Sialo lacto-N-tetraose

14 has a 4OH on terminal Galp, final capping might not be included.

S100: Chemical structure of compound 17. Please double check if internal Galp unit O6 position.

S101: Please double check if capping module is included or not during the Sia2a coupling. Due to heptasaccharide 17 has a 4OH on Galp at non reducing end. I believe that final capping step might not be included in the program.

S102: Please double check HRMS exp and found 17.

Concerns:

The authors display in the SI, ^{13}C - ^1H HMBC spectras for each one of the products after the automation and post automation products, to state anomeric α -configuration. To bring clarity towards the readers, can authors show an expansion of the spectra where it is highlighted the cross peak C1Sia-H3axSia, likewise Ando's showed for the compound 10 in the SM of 10.1126/science.aww4866)*... *at least for the products obtained by automation. Apart of this request, without being an expert in the sialic acid matters, this reviewer think if is that statement enough valid to ensure complete stereoselectivity & configuration? In other works from same authors, they display a 1D couple HMQC spectras or express $^3\text{J}_{\text{C}1\text{-H}3\text{ax}}$ values (likewise compound 12 in 10.3762/bjoc.11.69) or even selective proton decoupled ^{13}C -NMR is received well (likewise compounds 23 from 10.1021/acs.joc.9b01492 SI).

Please, see image attached.

Reviewer #3

(Remarks to the Author)

The manuscript by Kuo et al. outlines an automated process for synthesising human milk oligosaccharides (hMOs). The results concerning α 2,3- and α 2,6-sialylated hMOs are impressive, and the figures clearly depict the process and outcomes. Although my expertise lies in hMO bioactivity rather than organic chemistry, I was able to follow the experimental descriptions very well. Due to my lack of expertise in synthesis and NMR analysis, however, my comments are limited to the potential of the presented strategy for research and society.

Major:

Introduction:

- The authors do not mention HMOs at all in the introduction. I would suggest incorporating the first ten lines of the chapter 'Automated synthesis of sialylated HMOs' into the introduction.
- The authors should briefly specify the advantages of the automatic system in comparison to the aforementioned systems (purely chemical routes, enzymatic assembly, and chemoenzymatic methods): Speed, yield. This is not clear in the second paragraph of the introduction.

Results and Discussion:

- Please provide the quantity per chemical reaction approach for all synthesized hMOs and not only the yield if possible. However, the quantity is more important for users of HMOs than the yield. What quantities can be achieved with this automatic system for the respective hMO? The authors only refer to a "multi-gram scale". A table in the main section or in Figure 6 would be helpful for readers.
- The authors should briefly mention examples of the bioactivity of the respective hMOs that have been successfully synthesized.
- The authors should clearly state that most of the available data concerning the bioactivity of sialylated hMOs relates to sialyllactose, since it is widely available at low cost and can be tested in animal models or, subsequently, in clinical trials. Extensive studies of more complex structures have not been possible to date due to their expense. I see a potential advantage in the described strategy. The authors should discuss this critically.

Minor:

Line 3 of the introduction: Even though the N-glycosylation of RNA is still in its infancy, it should perhaps be mentioned in a subordinate clause.

Version 1:

Reviewer comments:

Reviewer #1

(Remarks to the Author)

The revised manuscript by Professor Seeberger and colleagues addresses most of the comments and suggestions

provided. The authors have highlighted the importance of solid-phase synthesis of sialylated glycans on the AGA platform rather than establishing a new glycosylation methodology or donor. The explanations provided by the authors are reasonable, given the complexity associated with refining a general method for the synthesis of higher-order glycans, especially those containing sialic acid moieties.

I am curious to know whether the authors are recovering and recycling the donor. I suggest that the authors describe these specific details in the Supporting Information and indicate the percentage of donors recovered, as this would be helpful for readers, particularly under solid-phase synthesis conditions on automated platforms. However, I do not believe this issue warrants further clarification in the main text.

I have no further suggestions, and I recommend accepting the article without further revision.

Reviewer #2

(Remarks to the Author)
Comments for the Authors

The concerns raised during review have been adequately addressed, and the requested corrections have been implemented.

The results are presented with excellent quality, and I believe the manuscript might meet the journal's standards.

Specifically, I am satisfied that the authors clarified that the requirement for an excess of sialic acid donors remains a limitation.

However, I would have appreciated a more explicit and detailed statement clarifying that the reported sialylation yields do not reflect the coupling step alone, but also incorporate the acid and pyridine washings, eventually a capping step, and the assumption that sialylated, non-sialylated, fucosylated, non fucosylated, 3O Lev, 3O Bz, and all previously capped glycans undergoes photocleavage with same efficiency when a comparison is assessed.

My only suggestion is to consider a slightly more informative title—for example, highlighting the number of complex sialylated HMOs synthesized via AGA, which in this work is nine. This is notable, given that among the approx. 200 known HMOs, sialylated HMOs represent only roughly 10–20% of the total.

I hope my comments have been helpful to the authors.

Reviewer #3

(Remarks to the Author)

The authors have responded very well to all suggestions for improvement, so in my opinion the manuscript is ready for publication.

Reviewer #1

1. Comment: The macrobicyclic sialic acid donors employed in this study have already been reported and extensively characterized by Kumura and co-workers in solution-phase synthesis. Simply transferring these previously established donors onto the AGA platform does not constitute sufficient novelty to attract the interest of Nat. Commun. readers.

Response: This macrobicyclic sialic acid donor was previously reported and characterized in solution-phase synthesis by Ando *et. al.* The novelty of our work is not the invention of the donor but establishing a general solid-phase synthesis of sialylated glycans on the AGA platform, supported by systematic screening, mechanistic insights, and demonstration on structurally demanding targets with an improved purification workflow.

- 1. Systematic AGA-specific conditions** (not derivable from solution): When we directly translated the reported solution-phase conditions to AGA environment, we observed reduced coupling efficiency. We attribute this to solid-support microenvironment effects (resin swelling, diffusion, local concentration, and donor stability). A key enabling finding is the introduction of a basic wash module, which suppresses donor-/acid-mediated degradation (Fig. S3).
- 2. Scope on challenging structures + actionable guidelines:** Using the optimized AGA-specific conditions, multi-mg scale automated synthesis of branched sialylated and fucosylated HMOs is feasible with simple routine operations. While doing so, we observed that remote protecting-group patterns and sequence order can strongly influence on-resin stability and coupling outcomes (Fig.4 & Fig.5). This finding will guide future building block design and synthetic sequence.
- 3. Improved global deprotection protocol:** We further demonstrated a global deprotection workflow that is compatible with both sialylated and fucosylated motifs, yielding products in high purity and useful quantities for downstream applications (mg range). This protocol and the AGA-specific conditions together remove a practical bottleneck for complex HMOs, which we believe is of broad interest to Nature Communications readers.

2. Comment: The yields reported for the sialylation steps remain low, despite the use of 20, 40, or even more equivalents of the sialic acid donor. Such high donor loadings with modest conversion raise concerns about the efficiency and robustness of the method.

Response: We agree that the use of excess donor is a current limitation of on-resin sialylation. However, the **isolated yields** reported in the manuscript reflect the full workflow (multi-step automated synthesis, photocleavage, and preparative HPLC recovery) and therefore should not be interpreted as a direct measure of the sialylation coupling efficiency alone. Importantly, the crude analytical HPLC/MS profiles show low to undetectable truncated/capped byproducts, consistent with efficient chain elongation under the optimized AGA conditions. We have also clarified in the manuscript that downstream recovery losses (photocleavage and preparative HPLC) materially contribute to the isolated yields. When desired, excess donor can be recovered and reactivated/recycled, which mitigates material use. In addition, commercial availability of the donor improves the practical accessibility of the method for broader use.

A similar concern was raised by Reviewer #2, Comment #6

Manuscript Updates: The statement was added: “The isolated yields were calculated based on purified oligosaccharides after full workflow and downstream recovery losses (photocleavage and preparative HPLC) over the theoretical maximum resin loading, and therefore should not be interpreted as a direct measure of the sialylation coupling efficiency alone.”

3. Comment: I am also curious whether, utilizing similar resources, the overall yields could be improved by employing disaccharide glycosyl donors that already contain the sialyl moiety. A brief discussion or comparison of this strategy would be helpful to contextualize the efficiency of the current approach.

Response: We acknowledge that employing disaccharide donors containing a pre-installed sialyl moiety is a valid strategy to reduce the number of coupling cycles and, in favorable cases, improve overall efficiency, an approach we had explored in the past (Figure 1a, Seeberger *et al. Beilstein J. Org. Chem.* **2012**, 8, 1601–1609). However, there are trade-offs between step economy and modularity: for complex branched targets such as DSLNF II (heptasaccharide **17**), the required branchpoint order and orthogonal protecting-group sequence favored stepwise assembly from monosaccharide building blocks followed by late-stage sialylation. More generally, pre-sialylated disaccharide donors can be advantageous for certain linear motifs, but they reduce flexibility for rapid sequence/pathway screening and library generation on AGA, whereas monosaccharide donors with orthogonal protecting groups preserve modularity and enable rapid optimization of synthetic sequences.

4. Comment: On page 2, left column, the authors state: “Solution-phase sialylation conditions developed by Ando *et al.* had to be adapted to solid-phase synthesis due to the physical and chemical difference between solution and solid phase synthesis approaches²⁶.” It is unclear which specific reference this statement is citing. The authors should clarify the corresponding citation.

Response: To prevent confusion between the method developed by Ando *et al.* and the discussion on solid-phase kinetics, reference 36 (Ando *et al. Science* **2019**, 364, 677-680) and reference 40 (Seeberger *et al. React. Chem. Eng.* 10, 1442-1454) are now cited.

Manuscript Updates: The sentence was revised: “Solution-phase sialylation conditions developed by Ando *et al.*³⁶ had to be adapted to solid-phase synthesis due to the physical and chemical difference between solution- and solid-phase synthesis approaches⁴⁰.”

5. Comment: Page 2, left column “Conversely, using too little dioxane compromised the stability of the activator system, resulting in variable yields (Entry 1-3).” The statement is misleading since the dioxane ratio is not too little in entries 1-3, its changing significantly in these three entries.

Response: Changes made

Manuscript Updates: The paragraph was revised: “Using a dichloromethane/dioxane mixture (4:1) for the activator solution (NIS/TfOH) gave the highest disaccharide conversion (Entry 2), whereas increasing the ratio of dioxane led to a slower reaction rate (Entry 1). Conversely, using **less** dioxane compromised the stability of the activator system, resulting in variable yields (Entry 3).”

6. Comment: Page 2, right column “A set of representative sialylated HMOs was targeted based on the lacto-N-tetraose (LNT) and the lacto-N-neotetraose (LNnT) tetrasaccharides core, to evaluate $\alpha(2,6)$ - and $\alpha(2,3)$ -sialylations at terminal and internal positions, as well as fucosylation in combination with sialylation”. Please refer to a scheme or provide a generic schematic representation for LNT or LNnT for the readers convenience.

Response: Changes are made.

Manuscript Updates: The paragraph was revised: “A set of representative sialylated HMOs was targeted based on the lacto-*N*-tetraose core (LNT, Gal $\beta(1,3)$ GlcNAc $\beta(1,3)$ Gal $\beta(1,4)$ Glc) and the lacto-*N*-neotetraose core (LNnT, Gal $\beta(1,4)$ GlcNAc $\beta(1,3)$ Gal $\beta(1,4)$ Glc), to evaluate $\alpha(2,6)$ - and $\alpha(2,3)$ -sialylations at terminal and internal positions, as well as fucosylation in combination with sialylation. In total, nine sialylated HMO targets were prepared (Fig. 3), featuring biologically relevant compounds such as the influenza decoy LSTc **9**^{45,46}, the NEC-preventing agent DSLNT **14**^{47,48}, and the cancer-associated glycan DSLNF II **17**^{49,50}.”

7. Comment: What are the isolated yields for the individual sialylation steps in the preparation of the mono- and disialylated oligosaccharides 9–14?

Response: We did not isolate intermediates after each sialylation step for compounds **9–14**, as the automated sequence was programmed to run till the final product; therefore, isolated yields for individual sialylation steps are not available. Overall step performance was instead monitored via crude analytical HPLC/MS after photocleavage, which shows minimal deletion byproducts, suggesting productive chain elongation throughout the programmed synthesis. When spectra of crude mixtures indicated incomplete conversion or significant byproduct formation, we then performed systematic follow-up optimization and mechanistic analysis, as exemplified by the development work required for DSLNF II (Fig. 5, compound **17**).

8. Comment: Is the overall yield of DSLNF II by AGA is significantly high compared to Kiso’s total synthesis?

Response: Our automated synthesis of DSLNF II (12% end-to-end isolated yield, 12 automated steps, ~21 hours instrument runtime, excluding downstream workup/purification). In comparison, Kiso’s method published in *Carbohydr. Res.* 2003, 338, 503-514 reports an overall yield of ~5% over 15 steps. While direct comparisons across platforms and protecting-group strategies are inherently approximate, we can assess that AGA route compares favorably in overall yield and, importantly, substantially reduces extensive manual handling and repeated purifications typical of multi-step solution-phase oligosaccharide synthesis. This method enables access to DSLNF II on a much shorter timeline and is reproduced consistently.

Reviewer #2

1. **Comment:** Page 2, left column. Add reference after “Ando et al”. (Probably 10.1126/science.aaw4866)

Response: Changes made

Manuscript Updates: The sentence was revised: “Solution-phase sialylation conditions developed by Ando *et al.*³⁶ had to be adapted to solid-phase synthesis due to the physical and chemical difference between solution- and solid-phase synthesis approaches⁴⁰.”

2. **Comment:** Fig 3. Right to the top: Please double check if products should include Sia ($\alpha 2 \rightarrow 6$) GlcN structure.

Response: Changes made

Manuscript Updates: Sia ($\alpha 2 \rightarrow 6$) GlcN structure was included in Figure 3.

3. **Comment:** Fig 3. Structure of fucose 7a is missing.

Response: Changes made

Manuscript Updates: Fucose 7a was included in Figure 3.

4. **Comment:** Page 5. Left column, Automated synthesis of DSLNF II. Reference routes A1, A2 and B1, in the text.

Response: Changes made

Manuscript Updates: Route A1, A2 and B1 were referred in section *Automated synthesis of DSLNF II (Fig. 5)*.

5. **Comment:** The excess equivalents of the bicyclic Sia donor that were needed show a weakness that this approach usually presents, and I think the authors could state this more explicitly. Ando’s stereoselective glycosylations in solution are excellent (83–99%) using just a 1.0:1.0 donor/acceptor ratio (10.1126/science.aaw4866). It is not clear to me if the yields obtained by the authors are inherent to sialylations performed in solid phase, or if Ando’s solution-phase glycosylations cannot be compared directly, because the efficiency of the cleavage of the sialylated products from the solid support either decomposes the sialoside or makes the cleavage more difficult. Could the authors please make comment on this?

Response: We agree with the reviewer that the requirement for excess bicyclic donor is a current limitation of on-resin sialylation, and we now state this more explicitly in Conclusion. The higher donor equivalents primarily reflect heterogeneous solid-phase kinetic constraints: at the low temperature used to suppress donor elimination, resin swelling and effective mass transfer are also reduced, and the reaction becomes diffusion-limited within the resin microenvironment. Under these conditions, donor consumption via competitive pathways (including elimination) can outcompete productive coupling unless donor is supplied in excess. Thus, high donor loading acts as kinetic compensation for on-resin sialylation.

Regarding the comparison to Ando’s solution-phase sialylations (Ando *et al. Science* **2019**, 364, 677–680), we note that coupling reactions are conducted in solution-phase and are highly sensitive to moisture content/conditioning (e.g., in Table S4 the authors note decreased yield in the absence of molecular sieves for related donors). In contrast, AGA is performed in solid-

phase synthesis where local micro-acidity, diffusion, and conditioning cannot be assumed to replicate solution-phase behavior even under nominally similar reagent ratios.

Manuscript Updates: The sentence was added in Conclusion: “The requirement for excess bicyclic donor is a current limitation, however, the excess donor can be recovered and reactivated/recycled, which helps mitigate material use.”.

6. Comment: It is understood that the yields shown in the tables and in all the Figures correspond to the moles of product obtained after purification, once the complete automated cycle of glycosylations, capping, tPG removal, and sialylations has been completed, and that this includes the post-automation cleavage process. With this in mind, I would like to emphasize that the yields reported cannot be directly or specifically associated with the actual coupling efficiency of the sialic acid. Such an association could only be made if the authors mentioned at least that they are assuming that all oligosaccharides, regardless of size or whether they are fuco or sialylated or not, cleave within a similar range. Therefore, I suggest to the authors mention this point somewhere in the manuscript in order to guard the analysis, towards the analysis & interpretation of the reported yields. Readers should acknowledge there are so many possible confounding variables when a tremendous complex work is undergoing that some supposition has to be made at the time of analyzing outcomes, likewise it is traditionally done by solution-phase carbohydrate chemists.

Response: The reviewer is correct in stating that the isolated yields reported in the manuscript reflect the full workflow (multi-step automated synthesis, photocleavage, and preparative HPLC recovery) and therefore should not be interpreted as a direct measure of the sialylation coupling efficiency alone. Importantly, the crude analytical HPLC/MS profiles show low to undetectable truncated/capped byproducts, consistent with efficient chain elongation under the optimized AGA conditions. We have also clarified in the manuscript that downstream recovery losses (photocleavage and preparative HPLC) materially contribute to the isolated yields.

A similar concern was raised by Reviewer #1, Comment #2

Manuscript Updates: The sentence was added: “The isolated yields were calculated based on purified oligosaccharides after full workflow and downstream recovery losses (photocleavage and preparative HPLC) over the theoretical maximum resin loading, and therefore should not be interpreted as a direct measure of the sialylation coupling efficiency alone.”

7. Comment: Page 4, left column. Whole paragraph that starts with “During the synthesis of hexasaccharide 15” until ends first paragraph, left column Page 5. This reviewer does not feel confident about the weight given to the discussion of the influence of the substituent at position 3 of GlcN on the yields. This is because it is not simply the variation of the substituent, or even the change of the GlcN BB (8a or 8c); rather, the order of incorporation of the Galp is also being changed, and in addition there are tPG replacement reactions occurring. This concern is associated with the Conclusions section, fourth sentence that relates sialylation with assembling order.

Response: The order of incorporating the Galp moiety remained identical across the discussed entries: it's the second-last coupling before sialylation (Figure. 4c and extended

version Figure S5). HPLC of crude mixture showed only the sialylated and non-sialylated product as single isomers, which means that the tPG replacement reactions was complete.

Regarding the concern in section Conclusion, we clarify that this statement is substantiated by our investigation of the branched DSLNF II heptasaccharide **17**. In these experiments, each synthetic sequence produced a unique outcome, rather than yield fluctuations. We found that only one specific sequence was successful to produce DSLNF II, while the rest failed. The assembly order was confirmed to be a decisive factor.

8. Comment: Page 5. Left column. Automated synthesis of DSLNF II. "...(α 2,6) sialylation occurred exclusively at the primary acceptor". I assume this is expected by the authors, but I am not sure if that regioselectivity will be clear for a broader range of readers. Please, can you provide a reference that discuss this matter?

Response: Reference 37: Kiso *et. al. Carbohydr. Res.* 2003, 338, 503-514 and Reference 53: Fukase *et. al. Bioorg. Med. Chem.* 18, 3760-3766 demonstrated selective α (2,6)-sialylation on 4,6-di-OH glucosamine acceptor.

Manuscript Updates: Reference 37 and 53 were cited: "After removing both Lev groups to expose the 4- and 6-hydroxyl groups, α (2,6)-sialylation occurred exclusively at the primary acceptor^{37,53}, followed by α (1,4)-fucosylation at 4-OH (Route A1)."

9. Comment: S6: Fig S5. Titles (A) and (C), and corresponding left structures, 9 and 18 might be reviewed. Compound 9 is a 3OBn protected GlcN. Compound 18 has a 3OBz on GlcN.

Response: Changes made

Manuscript Updates: The titles and corresponding structures in Fig.S5 were updated.

10. Comment: S11: Synthesis of S5. Ester reduction to alcohol shouldn't be with aluminum based hydrate (LiAlH₄) instead of NaBH₄? Please double check.

Response: Ester reduction was performed with NaBH₄, which is consistent with the procedure reported by Ando *et. al. Science* **2019**, **364**, 677-680.

11. Comment: S12: Protected sialic acid chemical structure. Should say compound S7 instead of S6. S6 is SM.

Response: Changes made

Manuscript Updates: The name was updated to **S7**.

12. Comment: S40: Levulinic anhydride synthesis. Please double check if no base was required here.

Response: The preparation of levulinic anhydride was prepared following the method reported by Patchornik *et. al. J. Am. Chem. Soc.* **1975**, 97, 1614-1615. No base was required.

Manuscript Updates: Reference 5 was added to the preparation of levulinic anhydride in S41 and S44.

13. Comment: S43: Synthesis of S16. Last paragraph: please double check if should say S16, instead of S15.

Response: Changes made

Manuscript Updates: The text was updated to **S16**.

14. Comment: S44: Synthesis of 8f. First paragraph: please double check if should say S16, instead of S15.

Response: Changes made

Manuscript Updates: The text was updated to **S16**.

15. Comment: S46: Please disclose the type and amount of resin used (might be mentioned elsewhere but this reviewer couldn't find it).

Response: The preparation of Merrifield resin was cited in Section 4.1 to Reference 6 (Seeberger *et al. J. Am. Chem. Soc.* **2019**, 141, 9079-9086). The amount of resin (40 mg) was mentioned in all tables in Section 5.

16. Comment: S55: All NMR spectra of samples prepared in CDCl₃ were collected at 50 oC, is there any specific/special reason behind?

Response: The macrobicyclic sialic acid building blocks and the corresponding protected oligosaccharides exhibit restricted conformational mobility and rotameric populations, which at room temperature can lead to slow exchange on the NMR timescale, resulting in line broadening and poorly resolved multiplets in CDCl₃. Collecting spectra at 50 °C, as also reported by Ando *et al. Science* 2019, 364, 677–680, improves spectral resolution (sharper resonances and more interpretable coupling patterns) and facilitates reliable assignment. We did not observe evidence of sample degradation during the NMR acquisition.

17. Comment: S72: Please double check if capping module is included or not during the Gal3b coupling. Due to Sialo lacto-N-neotetraose 11 has a 3OH on terminal Galp, final capping might not be included.

Response: Standard capping module was indeed included after Gal 3b coupling but before 3-Fmoc deprotection. Hence the protected oligosaccharide product has a free 3-OH.

18. Comment: S86: Please double check if capping module is included or not during the Sia2a coupling. Due to Di Sialo lacto-N-tetraose 14 has a 4OH on terminal Galp, final capping might not be included.

Response: Standard capping module was indeed included after final Sia 2a coupling but we observed no acetylation. This is perhaps due to steric hindrance of 4-OH of Galp.

19. Comment: S100: Chemical structure of compound 17. Please double check if internal Galp unit O6 position.

Response: Changes made

Manuscript Updates: The correct structure was updated.

20. Comment: S101: Please double check if capping module is included or not during the Sia2a coupling. Due to heptasaccharide 17 has a 4OH on Galp at non reducing end. I believe that final capping step might not be included in the program.

Response: Standard capping module was indeed included after final Sia 2a coupling but we observed no acetylation. This is perhaps due to steric hindrance of 4-OH of Galp.

21. Comment: S102: Please double check HRMS exp and found 17.

Response: The HRMS was checked, and deviation was 3.2 ppm.

22. Comment: The authors display in the SI, ¹³C-¹H HMBC spectras for each one of the products after the automation and post automation products, to state anomeric α -configuration. To bring clarity towards the readers, can authors show an expansion of the spectra where it is highlighted the cross peak C1Sia-H3_{ax}Sia, likewise Ando's showed for the compound 10 in the SM of 10.1126/science.aww4866)*... *at least for the products obtained by automation. Apart of this request, without being an expert in the sialic acid matters, this reviewer think if is that statement enough valid to ensure complete stereoselectivity & configuration? In other works from same authors, they display a 1D couple HMQC spectras or express ³J_{C1-H3_{ax} values (likewise compound 12 in 10.3762/bjoc.11.69) or even selective proton decoupled ¹³C-NMR is received well (likewise compounds 23 from 10.1021/acs.joc.9b01492 SI). Please, see image attached.}

Response: We thank the reviewer for this suggestion. The expansion of the HMBC spectra for deprotected glycans is provided in the Supplementary Information. Correlation of C1Sia-H3_{ax}Sia in 2D HMBC experiments is sufficient to assign anomeric configuration, as also reported by Matta *et. al. Chem. Eur. J.*, **2001**, 7, 356-367 and Ando. *et. al. Science* **2019**, 364, 677-680. This correlation is commonly used as a diagnostic indicator for assignment of the α -sialoside configuration.

Manuscript Updates: Expanded/annotated HMBC spectra highlighting the C1Sia-H3_{ax}Sia cross peak were added to Section 6 (Global Deprotection of Sialylated Glycans) of the Supplementary Information.

Reviewer #3

1. Comment: The authors do not mention HMOs at all in the introduction. I would suggest incorporating the first ten lines of the chapter 'Automated synthesis of sialylated HMOs' into the introduction.

Response: We thank the reviewer for this suggestion. We agree that the paragraph should be incorporated into Introduction.

Manuscript Updates: The paragraph was moved to the Introduction: "Human milk oligosaccharides (HMOs), found in human breast milk, is the third most abundant solid component of milk⁷ with key roles in infant nutrition, immune development, and gut microbiota modulation⁸⁻¹⁰. One fifth of the over 200 distinct HMO structures are sialylated¹¹⁻¹⁴. Access to pure molecules is crucial for elucidating their biological functions, yet natural sources yield only minute quantities that are extremely hard to isolate individually¹⁵."

2. Comment: The authors should briefly specify the advantages of the automatic system in comparison to the aforementioned systems (purely chemical routes, enzymatic assembly, and chemoenzymatic methods): Speed, yield. This is not clear in the second paragraph of the introduction.

Response: We thank the reviewer for this suggestion. We agree that the comparison should be incorporated into Introduction.

Manuscript Updates: The following were added:

- "While purely chemical synthesis offers great flexibility, it is labor-intensive and time-consuming due to the requirement for purification after each step. Enzymatic and chemoenzymatic approaches offer complete stereoselectivity but are often limited by the availability and substrate specificity of glycosyltransferases."

- "AGA platform bypasses intermediate purifications and automates the assembly process, significantly accelerating the production of HMOs from weeks or months to a matter of days²⁹."

3. Comment: Please provide the quantity per chemical reaction approach for all synthesized hMOs and not only the yield if possible. However, the quantity is more important for users of HMOs than the yield. What quantities can be achieved with this automatic system for the respective hMO?The authors only refer to a "multi-gram scale". A table in the main section or in Figure 6 would be helpful for readers.

Response: We clarify that the "multi-gram scale" referred to the preparation of building blocks. The automated syntheses were conducted on an analytical scale, typically yielding 1-2 mg of deprotected HMOs per single run. The fully automated nature of the platform allows for the straightforward accumulation of desired glycans by doubling/tripling resin and building block amount or simply repeating the synthesis. Detailed isolated quantities for each compound are indeed provided in the Supplementary Information.

4. Comment: The authors should briefly mention examples of the bioactivity of the respective hMOs that have been successfully synthesized.

Response: We thank the reviewer for this suggestion. We agree that highlighting the biological relevance of the synthesized targets significantly strengthens the manuscript. We

have updated the text to explicitly mention the bioactivities of key HMOs, supported by established literatures.

Manuscript Updates: The paragraph was revised to read: A set of representative sialylated HMOs was targeted based on the lacto-*N*-tetraose core (LNT, Gal β (1,3)GlcNAc β (1,3)Gal β (1,4)Glc) and the lacto-*N*-neotetraose core (LNnT, Gal β (1,4)GlcNAc β (1,3)Gal β (1,4)Glc), to evaluate α (2,6)- and α (2,3)-sialylations at terminal and internal positions, as well as fucosylation in combination with sialylation. In total, nine sialylated HMO targets were prepared (Fig. 3), featuring biologically relevant compounds such as the influenza decoy LSTc **9**^{45,46}, the NEC-preventing agent DSLNT **14**^{47,48}, and the cancer-associated glycan DSLNF II **17**^{49,50}.

5. Comment: The authors should clearly state that most of the available data concerning the bioactivity of sialylated hMOs relates to sialyllactose, since it is widely available at low cost and can be tested in animal models or, subsequently, in clinical trials. Extensive studies of more complex structures have not been possible to date due to their expense. I see a potential advantage in the described strategy. The authors should discuss this critically.

Response: We thank the reviewer for this suggestion. The current understanding of sialylated HMO bioactivity is indeed heavily towards simple structures like 3'-SL and 6'-SL due to their commercial availability.

Manuscript Updates: The sentence was added to Introduction: "To date, studies on sialylated HMOs have focused on simple trisaccharides, such as 3'-sialyllactose (3'-SL) and 6'-sialyllactose (6'-SL), primarily due to their commercial availability¹⁶⁻¹⁸."

6. Comment: Line 3 of the introduction: Even though the N-glycosylation of RNA is still in its infancy, it should perhaps be mentioned in a subordinate clause.

Response: Change made.

Manuscript Updates: The sentence was revised: "Sialic acids constitute a family of over 50 structurally distinct nine-carbon nonulosonic acid sugars¹, commonly found capping the non-reducing ends of glycans on glycoproteins, glycolipids and glycoRNAs²."

Reviewer #1

1. Comment: I am curious to know whether the authors are recovering and recycling the donor. I suggest that the authors describe these specific details in the Supporting Information and indicate the percentage of donors recovered, as this would be helpful for readers, particularly under solid-phase synthesis conditions on automated platforms.

Response: After solid-phase coupling, the reaction mixture can be recovered using the available fraction collector of the Glycoener® synthesizer. From this the major 2,3-eliminated byproduct any excess donor can be purified. An efficient method to transform this byproduct to a suitable building block is under investigation.

Reviewer #2

1. Comment: I would have appreciated a more explicit and detailed statement clarifying that the reported sialylation yields do not reflect the coupling step alone, but also incorporate the acid and pyridine washings, eventually a capping step, and the assumption that sialylated, non-sialylated, fucosylated, non fucosylated, 3O Lev, 3O Bz, and all previously capped glycans undergoes photocleavage with same efficiency when a comparison is assessed.

Response: The statement was added to the manuscript.

Manuscript Updates: “Isolated yields were determined for the fully purified oligosaccharides after completion of the entire workflow, including acidic wash, glycosylation, capping, deprotection, and material losses incurred during downstream purification steps such as photocleavage and preparative HPLC, relative to the theoretical maximum loading of the solid support. Because these values assume uniform efficiency across all intermediate steps, they do not directly reflect the coupling efficiency of the sialylation.”

2. Comment: Consider a slightly more informative title—for example, highlighting the number of complex sialylated HMOs synthesized via AGA, which in this work is nine. This is notable, given that among the approx. 200 known HMOs, sialylated HMOs represent only roughly 10–20% of the total.

Response: We thank the reviewer for highlighting the significance of synthesizing nine complex sialylated HMOs, which we have emphasized in the abstract and introduction section. We would prefer to retain the original title, as we feel it provides a concise and broad overview of the methodology and core message of the manuscript.